# Exploration of key markers driving ferroptosis in the progression of non-alcoholic fatty liver disease

**Jiyong Zhang**(ID), **Weiyan Li, Xiaoying Qiu, Yuanyuan Wang, Yongmei Zeng**(ID)*

Shenzhen Maternity and Child Healthcare Hospital, Women and Children's Medical Centre, Southern Medical University, Shenzhen, Guangdong Province, China

* zymdoctor2006@126.com

## Abstract

Non-alcoholic fatty liver disease (NAFLD) is a prevalent condition strongly linked to obesity, diabetes, and metabolic syndrome. Its global incidence is steadily increasing, placing a significant health burden on both patients and society. This study aims to identify ferroptosis-related differentially expressed genes (DEGs) through bioinformatics analysis and to explore their roles in NAFLD. By comparing samples from NAFLD patients and healthy controls, we identified 1,770 significant DEGs, with 1,073 being upregulated and 697 downregulated. Pathway analysis revealed a marked decrease in expression within certain key metabolic pathways (such as the one-carbon pool by folate) in the NAFLD group, while expression in DNA repair-related pathways (such as non-homologous end joining) was significantly increased. Additionally, immune cell infiltration analysis showed significant differences in 19 immune cell types between the NAFLD and control groups, with 12 types exhibiting increased infiltration in the NAFLD group. Through protein-protein interaction (PPI) network analysis, we identified 41 critical intersecting genes, and ROC curve validation demonstrated that 25 of these genes had an AUC value exceeding 0.85, highlighting their potential as biomarkers for NAFLD in early diagnosis and personalized treatment in the future.This study identifies critical ferroptosis-related genes and immune cell infiltration differences in NAFLD, offering potential biomarkers for early diagnosis and personalized treatment.

## Introduction

Non-alcoholic fatty liver disease (NAFLD) is a prevalent condition characterized by the accumulation of fat in the liver. It is closely associated with obesity, diabetes, and metabolic syndrome. The global prevalence of NAFLD is rising annually, making it a major public health challenge. This disease imposes significant health burdens and economic costs on both patients and society. Current diagnostic and therapeutic approaches for NAFLD primarily include lifestyle interventions and pharmacological treatm[ents. However,

**Data availability statement:** All data were obtained from the Gene Expression Omnibus (GEO) database (https://www.ncbi.nlm.nih.gov/geo/). The complete genome expression profiles for NAFLD were retrieved and downloaded using the "GEOquery" R package. The dataset GSE130970 includes samples from 74 NAFLD patients and 4 controls, while GSE89632 includes samples from 19 NAFLD patients and 24 controls. The data for ferroptosis-related genes were retrieved from the FerrDb database (http://www.zhounan.org/ferrdb/current/). This dataset comprises a total of 484 genes that were identified and curated in a previous landmark study (PMID: 32219413).

**Funding:** This work was supported by [Supported by Sanming Project of Medicine in Shenzhen, Grant Number SZSM202311021]. The funder had no role in study design, data collection and analysis, decision to publish, or preparation of the manuscript.

**Competing interests:** The authors have declared that no competing interests exist.

the lack of effective biomarkers and clear treatment strategies leaves many patients at the risk ofdisease progression. NAFLD not only affects the quality of life but could also lead to severe outcomes such as liver fibrosis, cirrhosis, and hepatocellular carcinoma [1].

In recent years, research on the mechanisms of NAFLD has grown substantially, and multiple studies have shown that dysregulation of iron metabolism is closely associated with its progression.NAFLD Patients are often accompanied by hepatic iron overload and elevated serum ferritin levels [2].Ferroptosis, a novel form of cell death, might play a crucial role in the development and progression of NAFLD.Ferroptosis is characterized by iron-dependent lipid peroxidation, Iron overload generates a large amount of reactive oxygen species (ROS) through the Fenton reaction, leading to mitochondrial membrane lipid peroxidation and disruption of cell membrane integrity,which aligns with the pathological mechanisms of NAFLD [3]. Existing studies suggested that ferroptosis might influence NAFLD by affecting cellular redox status and lipid metabolism, but the precise mechanisms remain unclear [4,5].

Despite emerging evidence linking ferroptosis to NAFLD progression, critical gaps persist in mechanistic. We aimed to identify novel biomarkers for detecting NAFLD progression. This study aims to bridge the gap in understanding the link between ferroptosis and NAFLD by employing bioinformatics analysis to identify differentially expressed genes related to ferroptosis in NAFLD patients. The approach integrates gene expression profiling and co-expression network analysis to explore the molecular mechanisms underlying NAFLD, potentially uncovering new diagnostic and therapeutic targets [6,7]. Investigating the relationship between NAFLD and ferroptosis might improve our understanding of NAFLD pathogenesis, enhance diagnostic accuracy, and contribute to the development of personalized treatment strategies, addressing a critical need in clinical research and treatment.

## Materials and methods

### Data collection

All data were obtained from the Gene Expression Omnibus (GEO) database (https://www.ncbi.nlm.nih.gov/geo/). The complete genome expression profiles for NAFLD were retrieved and downloaded using the "GEOquery" R package. The dataset GSE130970 includes samples from 74 NAFLD patients and 4 controls, while GSE89632 includes samples from 19 NAFLD patients and 24 controls. Batch correction was performed using the ComBat method in the "sva" R package [8]. Principal component analysis (PCA) was employed to examine the correction effect.In this study the data for ferroptosis-related genes were retrieved from the FerrDb database (http://www.zhounan.org/ferrdb/current/). This dataset comprises a total of 484 genes that were identified and curated in a previous landmark study (PMID: 32219413) [9].

### Ethics statement

All data was accessed on December 5th, 2023 from public database and was fully anonymized with no identifying patient information. Thus, informed consent and ethics review were waived by Ethics Committee of the Shenzhen Maternity and Child Healthcare Hospital.

## Weighted Gene Co-expression Network Analysis (WGCNA)

The WGCNA algorithm was implemented by using the WGCNA R package (version 1.70–3) to construct a co-expression network [10]. Pearson correlation analysis was used to calculate the association between module eigengenes (ME, the first principal component representing the overall expression level of a module) and ferroptosis, identifying modules significantly associated with ferroptosis. The co-expression module structure was visualized using a heatmap of gene network topological overlap. A hierarchical clustering tree and corresponding eigengene heatmaps were drawn to summarize the relationships between modules. Ferroptosis-related differentially expressed genes (ferroptosis-related DEGs) were obtained by intersecting DEGs with ferroptosis-related module genes.

## Differential expression analysis

Gene expression profiles were constructed for each sample from NAFLD patients and controls. The "limma" R package (version 3.50.0) was used to conduct differential expression analysis for NAFLD patients and healthy individuals in the GSE130970 and GSE89632 datasets. A screening threshold of $P < 0.05$ was set to obtain DEGs between NAFLD patients and controls.

## Gene Set Variation Analysis (GSVA)

To investigate biological function differences between the control and NAFLD groups, the "c2.cp.kegg.v7.5.1.symbols" gene set from the MSigDB database (http://software.broadinstitute.org/gsea/msigdb) was used as the reference. The "GSVA" R package (version 1.42.0) was employed for GSVA. Results were visualized using the "pheatmap" R package (version 1.0.12). Additionally, 50 hallmark gene sets were downloaded from the MSigDB database as the reference gene set. The ssGSEA function in the GSVA package was used to calculate GSVA scores for each gene set across different samples. The "limma" package was used to compare GSVA score differences between the control group and the NAFLD group.

## Gene ontology and kyoto encyclopedia of genes and genomes pathway enrichment analysis

The "clusterProfiler" R package (version 4.2.2) was applied for GO annotation and KEGG pathway enrichment analysis of ferroptosis-related DEGs. A P-value < 0.05 was considered statistically significant for enrichment. The GeneMANIA website (http://genemania.org) predicts relationships between functionally similar genes and hub genes, including protein-protein, protein-DNA interactions, pathways, physiological and biochemical reactions, co-expression, and co-localization [11].

## Protein – protein interaction network analysis

A PPI network was constructed by using the Search Tool for the Retrieval of Interacting Genes (STRING) online database, selecting proteins with an interaction score greater than 0.7 [12]. Visualization of the PPI network was enhanced with Cytoscape software [13]. The top 100 genes were extracted from all 12 algorithms in CytoHubba and their intersection was taken [14]. The 12 CytoHubba algorithms include: Betweenness, Stress, Radiality, Eccentricity, Degree, DMNC, EPC, MCC, Closeness, MNC, ClusteringCoefficient, and BottleNeck. The Cytoscape plug-in CytoHubba was used to detect hub genes in the PPI network. GeneMANIA could predict relationships between functionally similar genes and hub genes.

## Diagnostic value of hub genes

Receiver Operating Characteristic (ROC) curves were created by using the "pROC" R package to determine the area under the curve (AUC) for screening feature genes and evaluating their diagnostic value [15]. AUC values typically range between 0.5 and 1, with values closer to 1 indicating better diagnostic performance.

## Immune infiltration analysis

Single-sample gene set enrichment analysis (ssGSEA), an extension of gene set enrichment analysis (GSEA), calculates separate enrichment scores for each sample and gene set [16]. Relative enrichment scores of each immune cell in each sample's gene expression profile were calculated using the Tumor and Immune System Interaction Database (TISIDB) (http://cis.hku.hk/TISIDB/index.php) [17]. Changes in immune cell infiltration levels in NAFLD and control group samples were plotted by using the "ggplot2" R package (version 3.3.6) [18].

## Statistical analysis

Statistical analysis was performed by using R software v4.1.2. The Spearman correlation test was used to infer correlations between two parameters. The Wilcoxon test was applied to compare differences between two groups, and the Kruskal-Wallis test was used for comparisons among three or more groups. A two-sided P-value less than 0.05 was considered statistically significant.

## Results

### Construction of weighted gene co-expression network and module identification

WGCNA was applied to study gene sets related to ferroptosis. Scale independence and mean connectivity analysis showed that when the minimum soft threshold (β) equals 6 (Fig 1A), the average connectivity approaches 0, and scale independence is greater than 0.85. Four co-expression modules were identified, with unrelated genes assigned to the gray module, which was ignored in the following research (Fig 1B). To study the relationships among modules and determine their relevance, associations with ME were performed. Characteristics of gene networks were plotted using dendrograms and heatmaps (Fig 1C). A heatmap represented the topological overlap in gene networks (Fig 1D). To understand the physiological significance of intramodular genes, we associated four MEs with ferroptosis and sought the most significant associations. Based on the heatmap of module-trait relationships (Fig 1E), genes clustered in the turquoise module (n = 2012) had the strongest positive correlation with ferroptosis (r = 0.43, P < 0.05). Therefore, our subsequent focus was on the turquoise module, as it is likely to indicate ferroptosis more accurately. Fig 1F shows a scatter plot of gene significance (GS) versus module membership (MM) for ferroptosis traits in the turquoise module. The MM and GS for ferroptosis (Fig 1F) exhibited a highly significant positive correlation (cor = 0.59, P < 0.05), indicating that the most important (central) elements of the turquoise module also tend to be highly correlated with ferroptosis traits.

### Differentially expressed genes associated with non-alcoholic fatty liver disease

A total of 1,770 differentially expressed genes (DEGs) were identified by comparing NAFLD samples with controls. These genes demonstrated statistically significant differences between the two groups (adjusted p-value < 0.05 and |Log2 fold change| > 0.5). In the NAFLD samples, 1,073 genes were upregulated, while 697 genes were downregulated. All DEGs were visualized using a volcano plot (Fig 2A). Additionally, the top 5 upregulated genes (*AKTIP*, *VPS45*, *WNT5A*, *APOL3*, *KRT222*) and the top 5 downregulated genes (*FOSB, SOCS3, MYC, ADAMTS1, FAM107A*) were displayed using a heatmap (Fig 2B). Through rank-sum testing, it was also found that the top ten genes showed significant differences in expression between the NAFLD samples and the controls (p < 0.05, Fig 2C). The intersection of DEGs and genes from the ferroptosis-related module resulted in the identification of 1,389 ferroptosis-related DEGs.

### Gene set variation analysis

To explore functional annotations in NAFLD, we conducted a GSVA (Gene Set Variation Analysis) to assess the relative expression differences of pathways between the two groups. The GSVA analysis enriched for several differentially expressed pathways, which were visualized with a heatmap. Compared to the control group, the NAFLD group showed

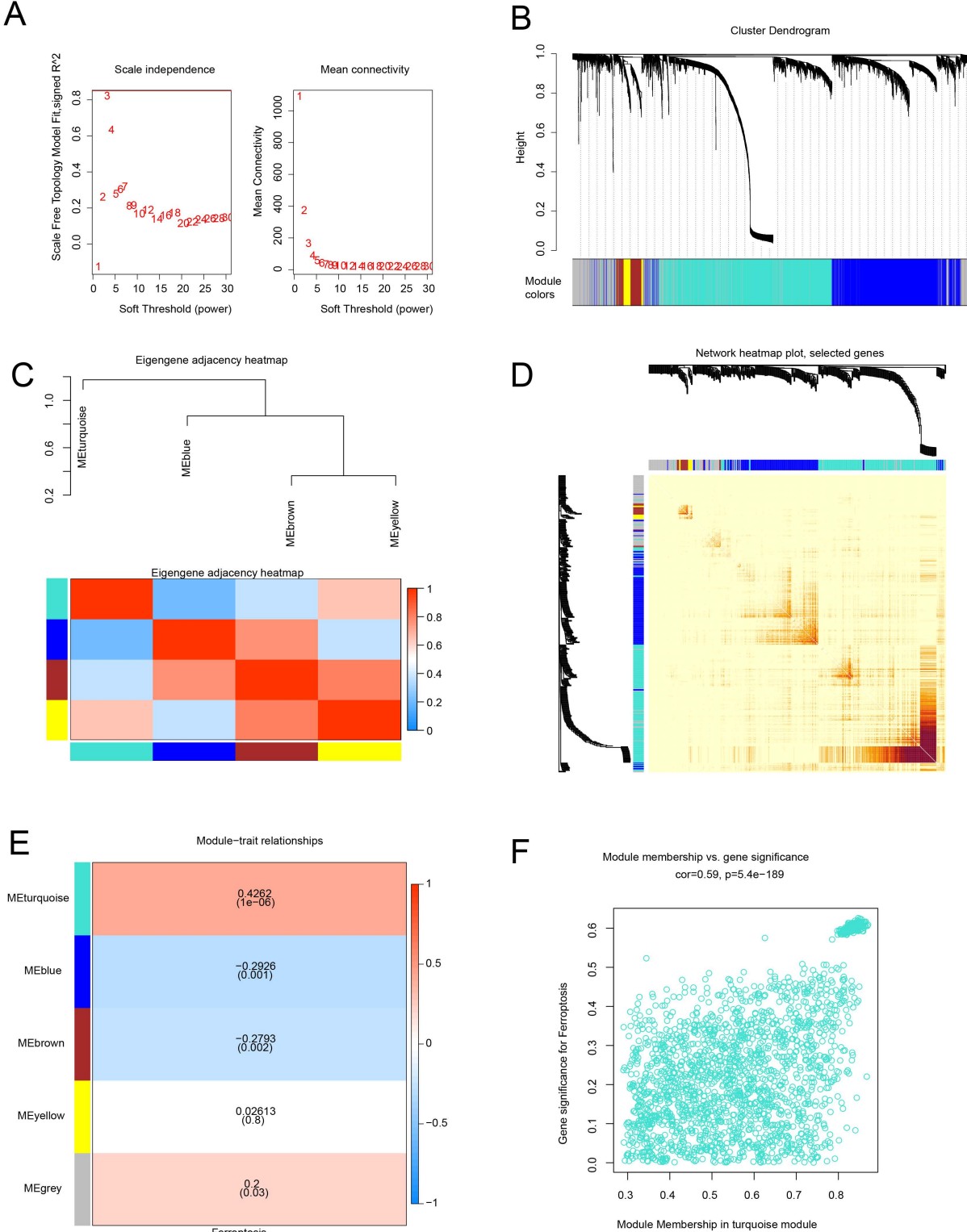

**Fig 1. Construction of the WGCNA Co-expression Network. (A)** Soft threshold β = 6, Scale-free Topology Fit Index (R2). **(B)** Analysis of the gene expression network in NAFLD identified different modules of co-expression data. **(C)** Relationships between modules. Top: Hierarchical clustering of module eigengenes summarizing the modules found in the clustering analysis. The branches (meta-modules) of the dendrogram combine eigengenes

with positive correlations. Bottom: Heatmap of correlations between module eigengenes. Each row and column in the heatmap corresponds to a module eigengene (marked by color). In the heatmap, red represents high adjacency, while blue indicates low adjacency. Red squares on the diagonal are meta-modules. **(D)** Heatmap of topological overlap in the gene network. In the heatmap, each row and column corresponds to a gene, with light colors indicating low topological overlap and increasingly darker reds indicating high topological overlap. Darker squares on the diagonal correspond to modules. The gene dendrogram and module assignment are shown on the left and top. **(E)** Relationships between consensus module eigengenes and ferroptosis. Each row in the table corresponds to a consensus module, and each column corresponds to a trait. The numbers in the table represent the correlation between the corresponding module eigengenes and traits, with P-values in parentheses printed below the correlations. Correlations are color-coded according to the color legend. **(F)** Correlation between MM* and GS* for ferroptosis-related genes in the turquoise module. Correpresents the absolute correlation coefficient between GS and MM. *: module membership (MM); gene significance (GS).

significantly lower expression in pathways such as One Carbon Pool by Folate and Glycosaminoglycan Biosynthesis Chondroitin Sulfate, while significantly higher expression was observed in pathways related to Non Homologous End Joining and Base Excision Repair (Fig 3).

### GO and KEGG pathway enrichment analysis

The GO results indicate that these genes are enriched in the following biological processes (BP): detection of chemical stimulus involved in sensory perception of smell (GO:0050911), sensory perception of smell (GO:0007608), and detection of chemical stimulus involved in sensory perception (GO:0050907). They are also enriched in cellular components (CC): intermediate filament (GO:0005882) and intermediate filament cytoskeleton (GO:0045111). For molecular functions (MF), enrichment was observed in olfactory receptor activity (GO:0004984), odorant binding (GO:0005549), and signaling receptor activator activity (GO:0030546) (Figs 4A and 4C). The enriched KEGG pathways include Olfactory transduction (hsa04740), Cytokine-cytokine receptor interaction (hsa04060), and Viral protein interaction with cytokine and cytokine receptor (hsa04061) (Figs 4B and 4D).

### PPI network

To further elucidate the highly connected genes within the PPI network, the intersection of ferroptosis-related DEGs was obtained using the CytoHubba algorithm, resulting in 41 intersecting genes. These include *JUN, BRCA1, EDN1, MYC, CCL5, FGFR2, PPARGC1A, IL6, ASPM, HSD3B1, CD247, CCNA2, HLA-C, CDT1, IL2, SDC4, IFNA1, TYMS, BUB1, OXT, SDC1, CDC20, CYP7A1, IL2RB, CXCR4, CXCL9, PBK, OAS2, RGS1, FPR1, IL4R, RAD51AP1, UBE2T, CD3E, CD3G, CD3D, TLR3, IFIH1, MND1, ISG15,* and *MCM5* (Fig 5).

### Diagnostic value of hub genes

To further validate the diagnostic value of the intersection genes, we used ROC curves to evaluate them and found that 25 genes had an area under the curve (AUC) greater than 0.85. These genes are *MYC* (AUC=0.928), *RAD51AP1* (AUC=0.922), *SDC1* (AUC=0.915), *PPARGC1A* (AUC=0.912), *TYMS* (AUC=0.908), *CDC20* (AUC=0.9), *UBE2T* (AUC=0.896), *CDT1* (AUC=0.895), *RGS1* (AUC=0.892), *ASPM* (AUC=0.891), *FGFR2* (AUC=0.89), *TLR3* (AUC=0.89), *BUB1* (AUC=0.887), *IL2RB* (AUC=0.876), *CD247* (AUC=0.871), *CYP7A1* (AUC=0.867), *PBK* (AUC=0.864), *SDC4* (AUC=0.864), *CXCL9* (AUC=0.864), *ISG15* (AUC=0.861), *JUN* (AUC=0.861), *EDN1* (AUC=0.859), *IL6* (AUC=0.859), *MND1* (AUC=0.858), and *CD3G* (AUC=0.854) (Fig 6,7). This indicates that these 25 genes have potential discriminatory power as biomarkers for non-alcoholic fatty liver disease, and they are considered hub genes in this study. Results for other genes are provided in the supplementary figure.

### Construction of weighted gene co-expression network and module identification

We used the GeneMANIA database to create a PPI network for the hub genes, where 25 genes showed interaction relationships (Fig 8A). We conducted GO and KEGG analyses on a total of 45 genes, including 25 hub genes and 20

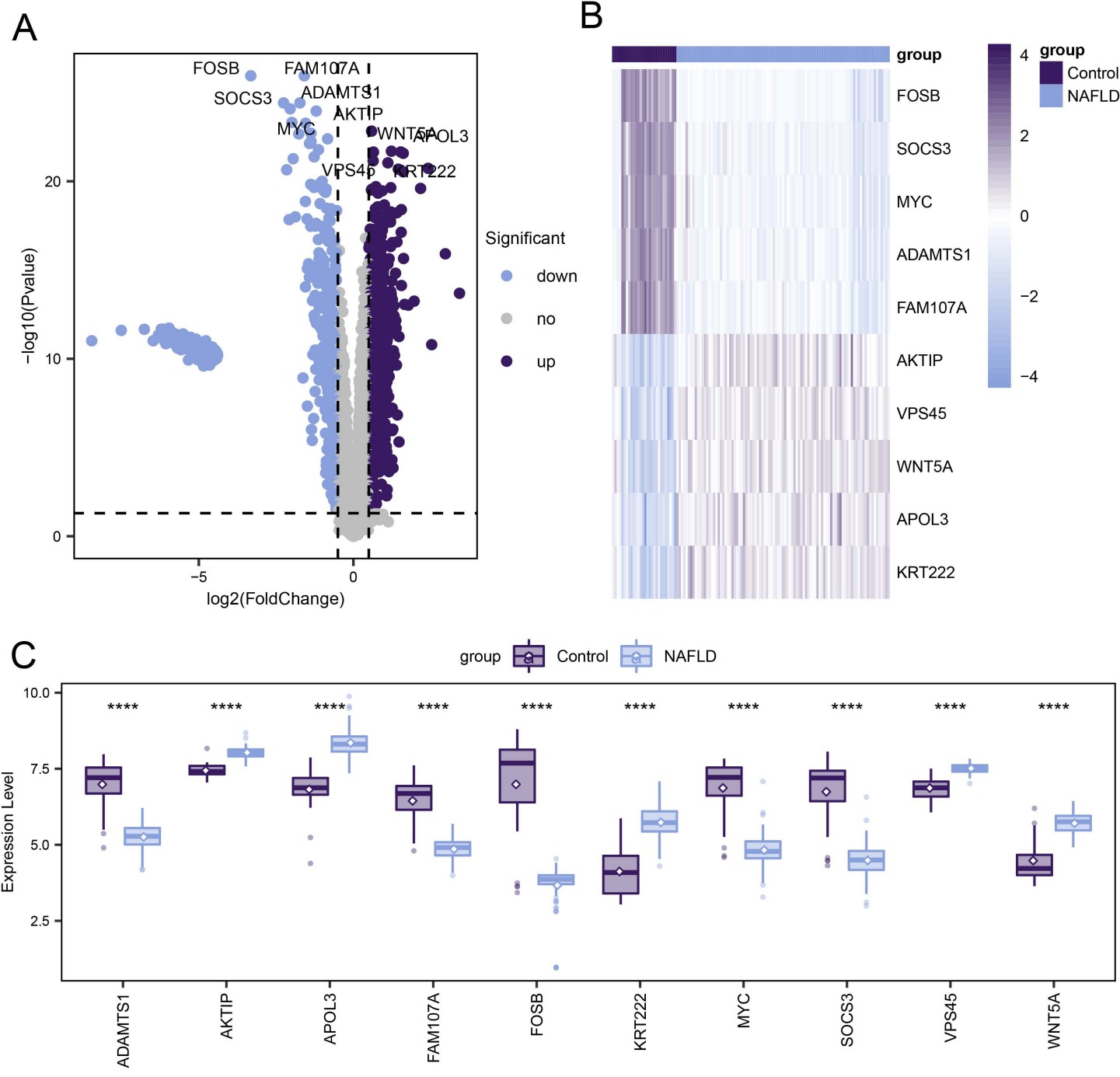

**Fig 2. Differential Gene Expression Associated with Non-Alcoholic Fatty Liver Disease. (A)** The volcano plot depicts the distribution of DEGs between NAFLD and control group samples. Gray dots represent genes with no significant differential expression levels. **(B)** The heatmap illustrates the top 5 upregulated and the top 5 downregulated DEGs. **(C)** The box plot shows differences in gene expression levels between NAFLD and control group samples, with statistical significance assessed using rank-sum testing. Asterisks indicate p-values: ****p < 0.0001, ***p < 0.001, **p < 0.01, *p < 0.05.

genes related to the hub genes. The GO enrichment results identified significantly enriched pathways such as metaphase plate congression, mitotic spindle assembly, regulation of morphogenesis of an epithelium, and regulation of the stress-activated MAPK cascade (Fig 8B).

The KEGG analysis showed that the main enriched pathways included Human T-cell leukemia virus 1 infection, Cell cycle, Epstein-Barr virus infection, and Th17 cell differentiation (Fig 8C). By investigating the gene expression of key pathways, we verified that there were significant differences between the disease group and the control group in gene sets related to tumor necrosis factor superfamily cytokine production and the PI3K−Akt signaling pathway (Figs 9A and 9B).

## Immune infiltration

Immune cell infiltration might play a crucial role in the pathogenesis of non-alcoholic fatty liver disease (NAFLD). Therefore, we investigated the association between NAFLD and infiltrating immune cells in control samples. Among the 28 types of immune cells, 19 showed significant differences in immune infiltration abundance between the two groups (p < 0.05) (Fig 10A). Twelve of these immune cells had significantly higher levels of infiltration in the NAFLD group compared to the control group (Fig 10A). As shown in Fig 10B, the overall level of immune cell infiltration differed greatly between the NAFLD and control groups. We also examined the significant correlation between each hub gene and the corresponding immune cells. IL6 was significantly correlated with Neutrophils (R = 0.697, p < 0.001) (Fig 10C), and IL6 was also significantly correlated with Eosinophils (R = 0.77, p < 0.001) (Fig 10D). Additionally, we tested for correlations among immune cells and found a generally positive correlation (Fig 10E).

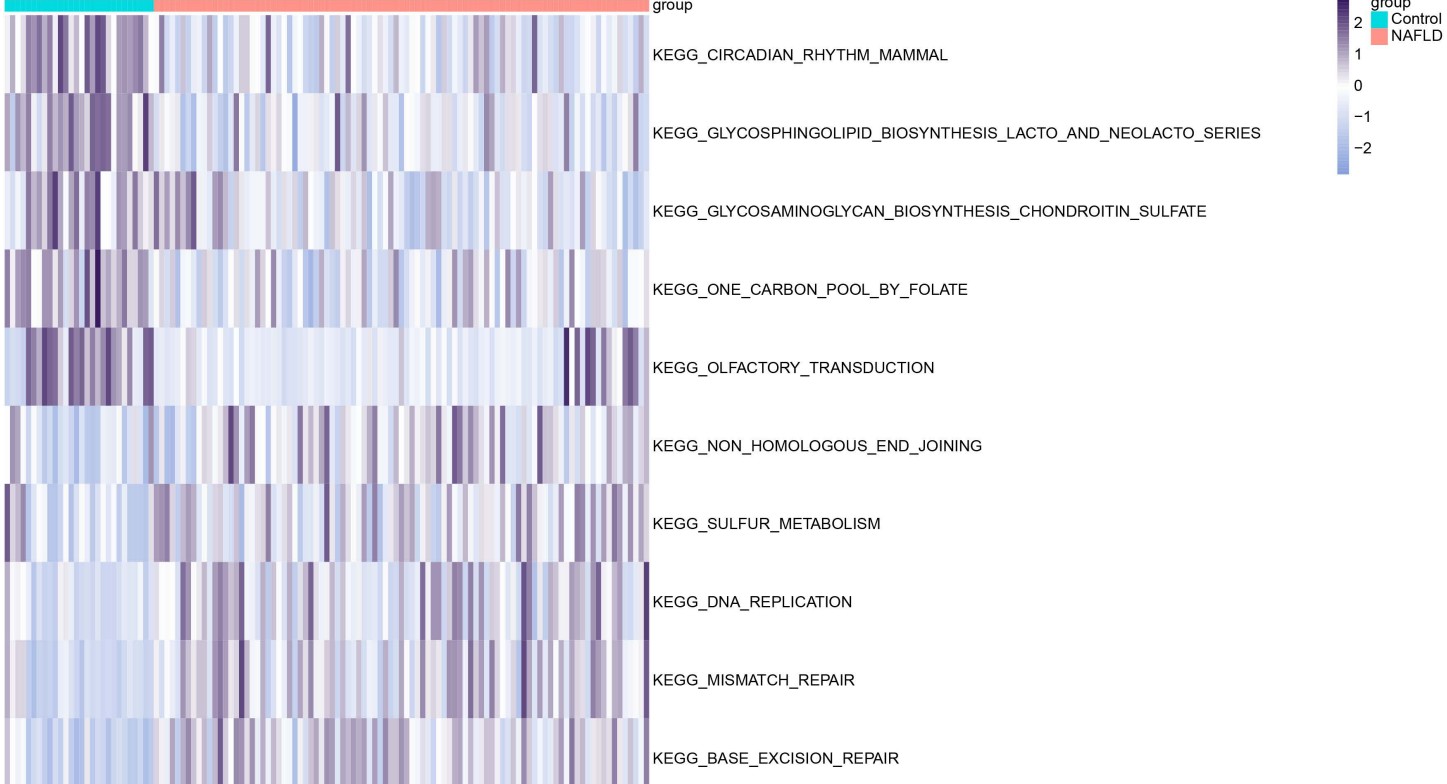

**Fig 3. Significantly Enriched Pathways.** Visualization of GSVA analysis through a heatmap.

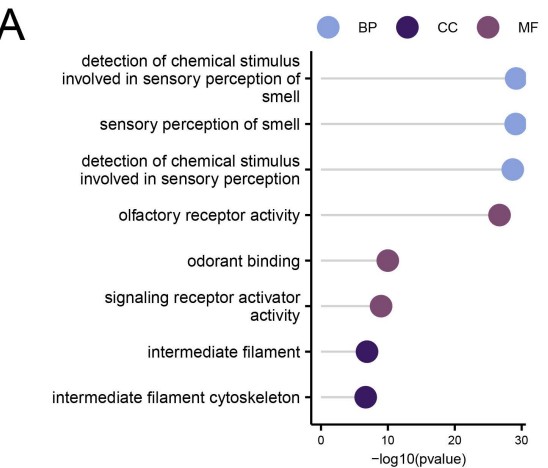

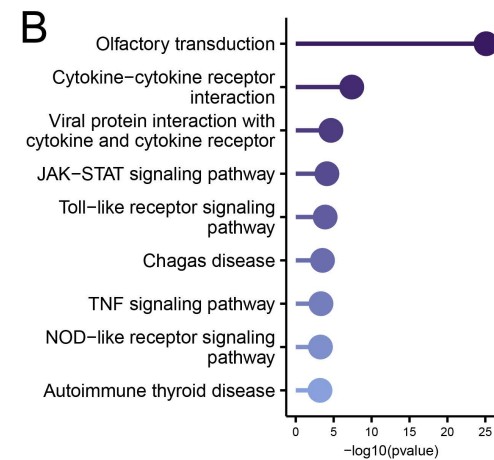

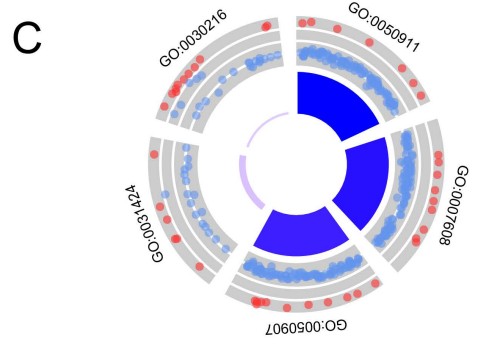

| ID | Description |
|---|---|
| GO:0050911 | detection of chemical stimulus involved in sensory perception of smell |
| GO:0007608 | sensory perception of smell |
| GO:0050907 | detection of chemical stimulus involved in sensory perception |
| GO:0031424 | keratinization |
| GO:0030216 | keratinocyte differentiation |

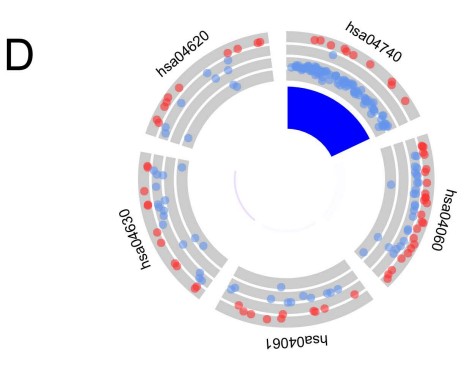

| ID | Description |
|---|---|
| hsa04740 | Olfactory transduction |
| hsa04060 | Cytokine–cytokine receptor interaction |
| hsa04061 | Viral protein interaction with cytokine and cytokine receptor |
| hsa04630 | JAK–STAT signaling pathway |
| hsa04620 | Toll–like receptor signaling pathway |

**Fig 4. Enrichment Analysis Based on Ferroptosis-Related Differentially Expressed Genes. (A)** GO term enrichment analysis results for ferroptosis-related differentially expressed genes. **(B)** KEGG pathway enrichment analysis results for ferroptosis-related differentially expressed genes. **(C)** Circle plot displaying the correspondence between enriched GO terms and pathway names. **(D)** Circle plot displaying the correspondence between enriched KEGG pathways and pathway names.

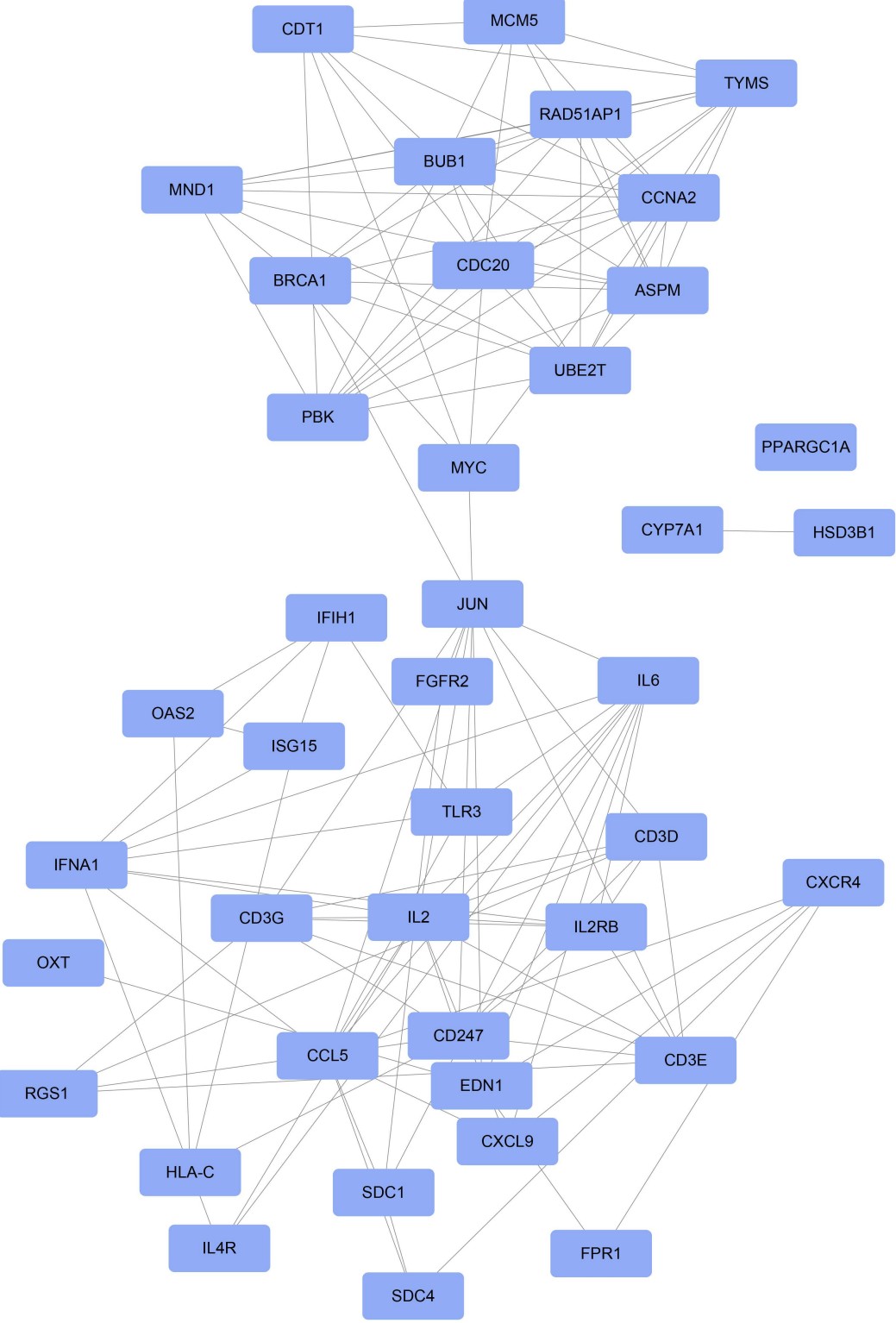

**Fig 5. Intersection genes obtained from the PPI network and PPI network analysis of intersection genes. colored circles represent genes.**

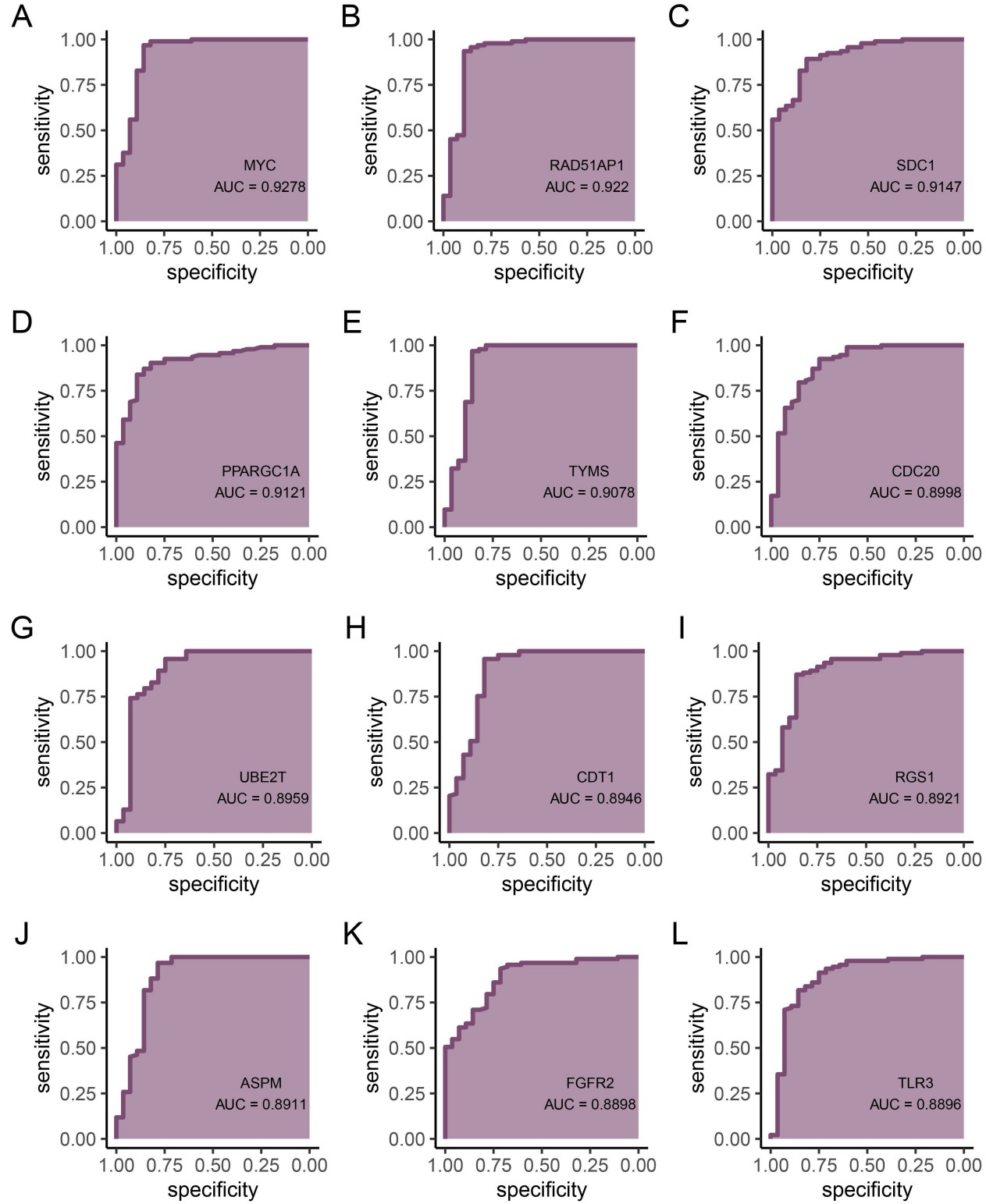

**Fig 6. ROC curves for hub genes. (A)** *MYC*. **(B)** *RAD51AP*1. **(C)** *SDC1*. **(D)** *PPARGC1A*. **(E)** *TYMS*. **(F)** *CDC20*. **(G)** *UBE2T*. **(H)** *CDT1*. **(I)** *RGS1*. **(J)** *ASPM*. **(K)** *FGFR2*. **(L)** *TLR3*.

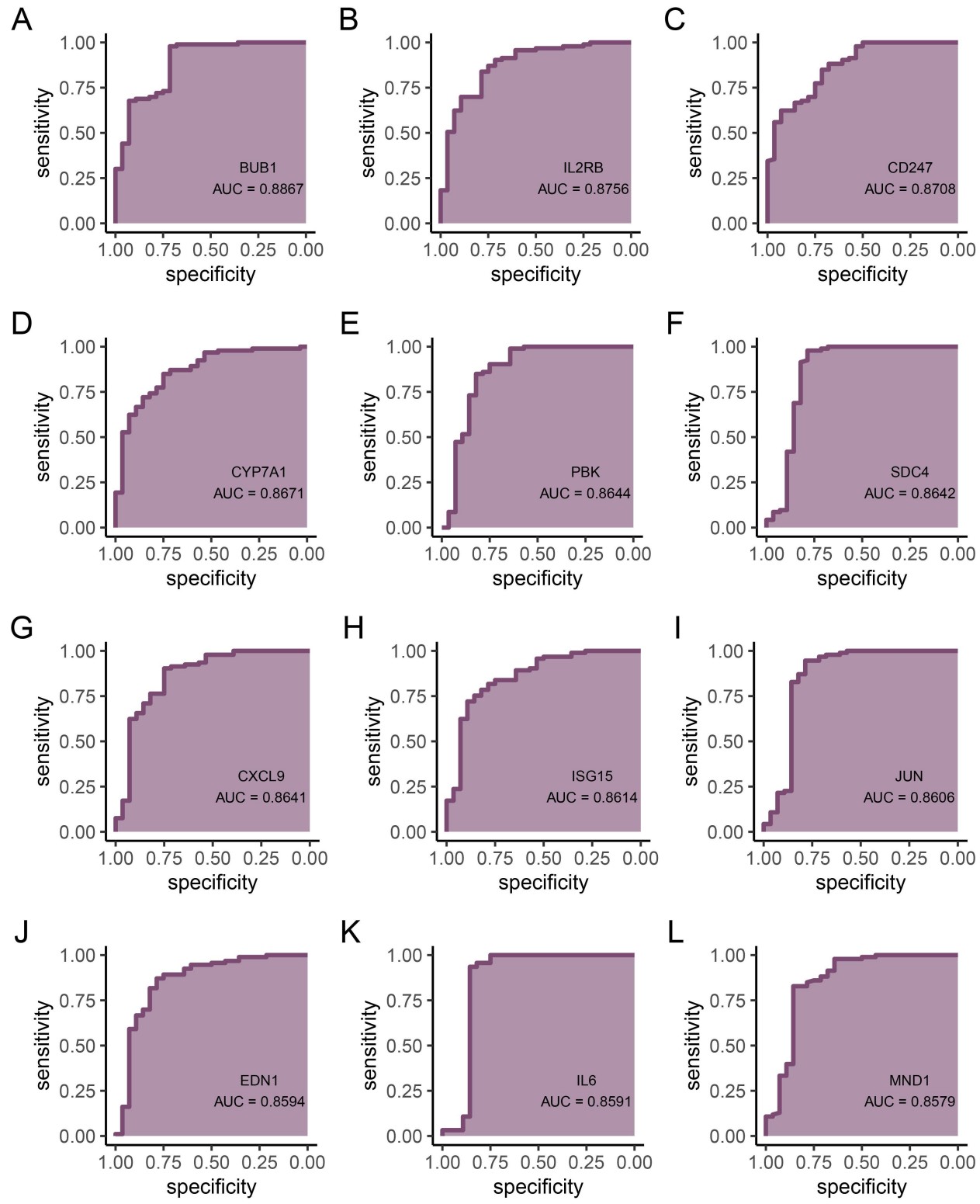

**Fig 7. ROC curves for hub genes. (A)** *BUB1*. **(B)** *IL2RB*. **(C)** *CD247*. **(D)** *CYP7A1*. **(E)** *PBK*. **(F)** *SDC*4. **(G)** *CXCL*9. **(H)** *ISG*15. **(I)** *JUN*. **(J)** EDN1. **(K)** *IL*6. **(L)** *MND*1. **(H)** *CD*3G.

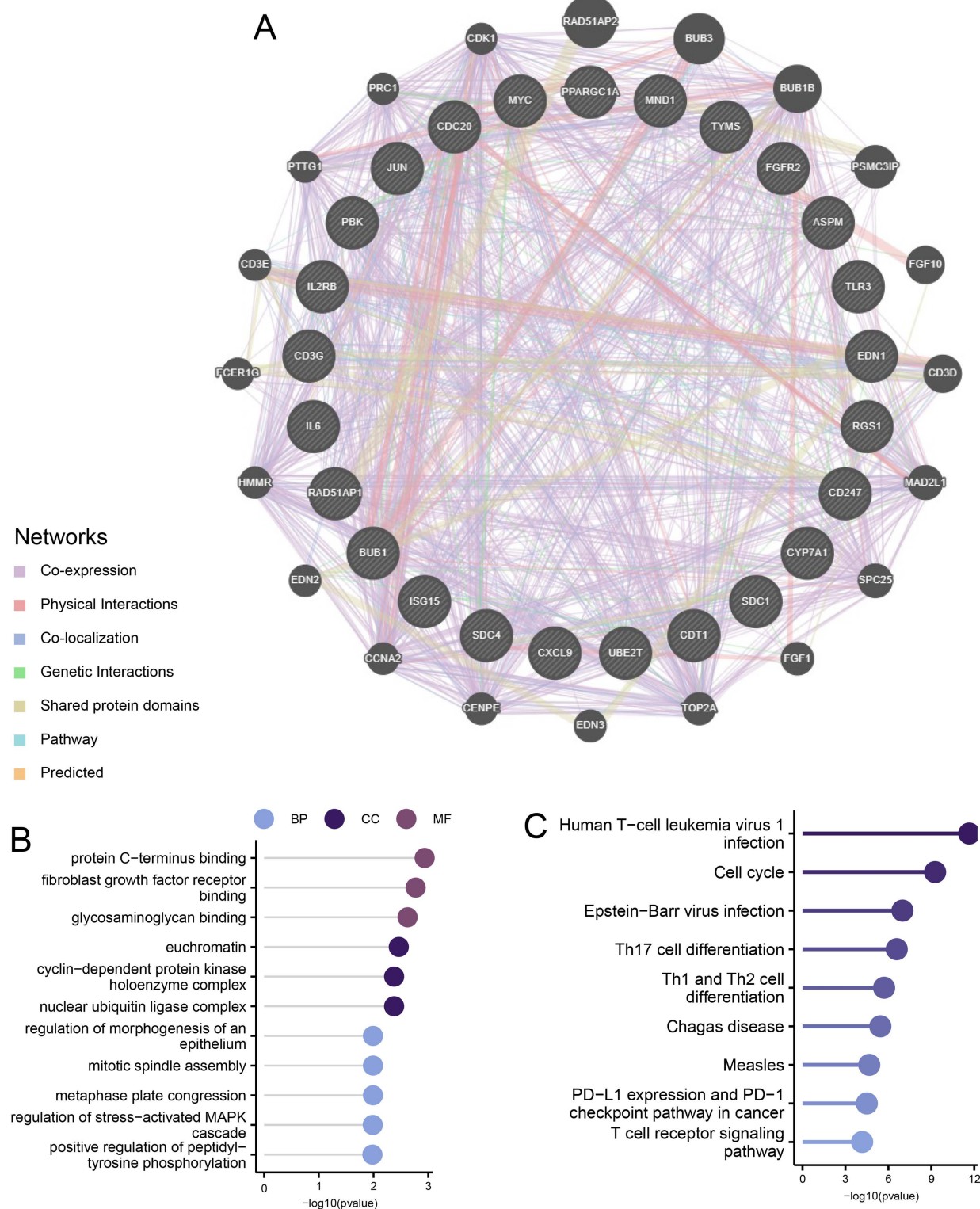

**Fig 8. Interaction analysis of hub genes. (A)** Gene Co-expression Network Diagram. **(B)** GO Analysis of Co-expressed Genes. **(C)** KEGG Analysis of Co-expressed Genes.

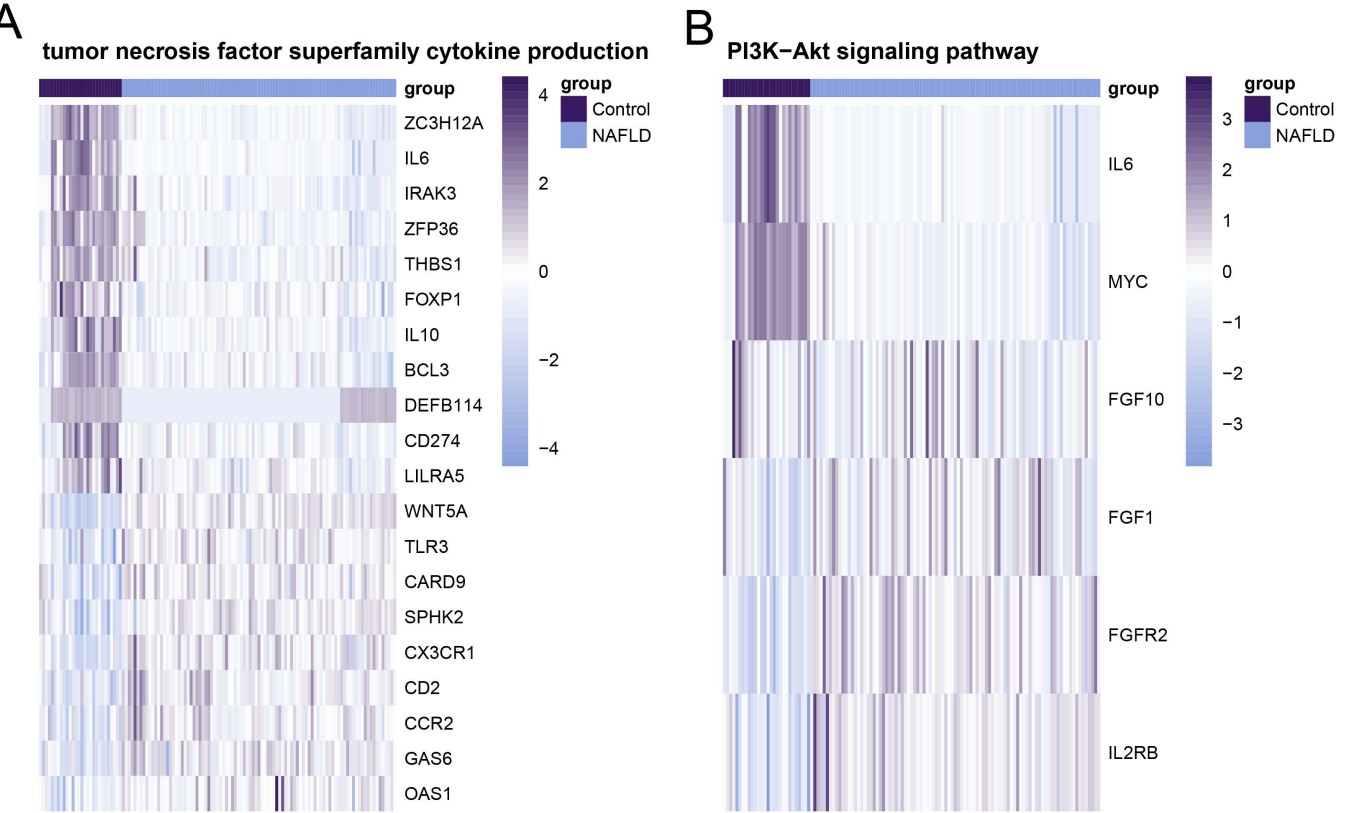

**Fig 9. Expression profile heatmaps of key genes in gene sets. (A)** Tumor necrosis factor superfamily cytokine production in non-alcoholic fatty liver disease and control samples. **(B)** PI3K−Akt signaling pathway in non-alcoholic fatty liver disease and control samples.

## Pathways related to key genes

Using GSVA, we further studied the differences in 50 hallmark signaling pathways between NAFLD patients and control groups. In NAFLD patients, 13 hallmark signaling pathways were significantly upregulated, while 14 pathways were significantly downregulated (Fig 11A). We also analyzed the correlation between the 5 most significantly differentially expressed hub genes and the 50 hallmark signaling pathways. *MYC* was found to be associated with many pathways, including HALLMARK_UV_RESPONSE_UP and HALLMARK_UV_RESPONSE_DN (Fig 11B).

## Discussion

NAFLD is the most common chronic liver disease globally, with a prevalence rate of 25%−38% worldwide, particularly high in Latin America and the Middle East (prevalence rates of 44.37% and 36.53%, respectively) [19]. With the rising incidence of obesity and metabolic syndrome, the number of NAFLD patients is expected to exceed 350 million by 2030. Approximately 30% of NAFLD patients progress to non-alcoholic steatohepatitis (NASH), with 15%−20% ultimately developing cirrhosis or hepatocellular carcinoma (HCC) [20]. Currently, the treatment of NAFLD primarily includes lifestyle interventions and medications (such as vitamin E, GLP-1 agonists); however, these approaches suffer from poor long-term adherence and significant side effects (including bleeding and gastrointestinal reactions) and mainly improve early pathologies (like steatosis) with limited effectiveness on fibrosis and cirrhosis [21,22]. Research on ferroptosis offers a new direction to address this challenge. Ferroptosis inhibitors (such as Ferrostatin-1 and quercetin) can alleviate liver

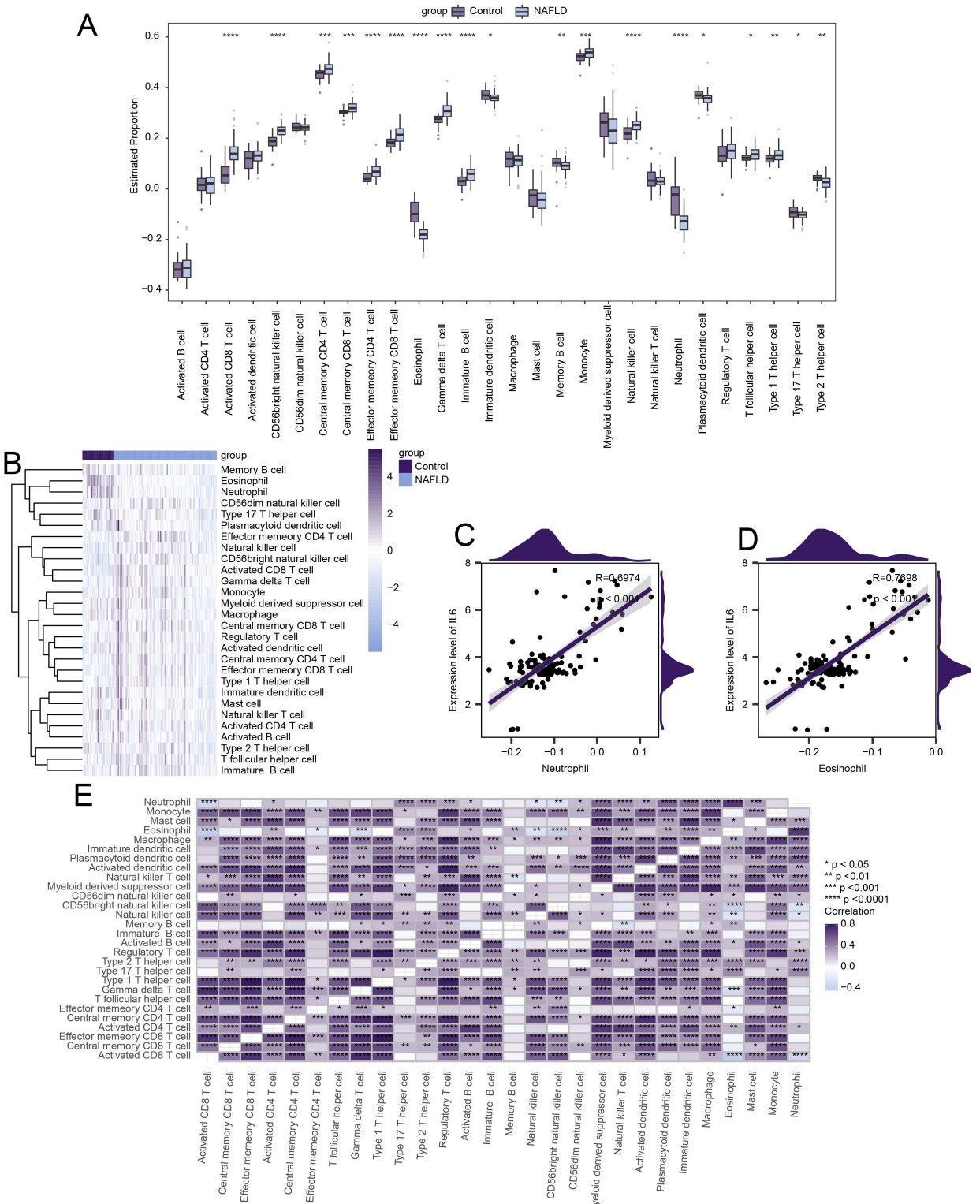

**Fig 10. Differences in immune infiltration between NAFLD and control groups. (A)** Heatmap of changes in immune infiltration levels between NAFLD and control groups. **(B)** Differences in the estimated proportions of immune cell infiltration between the NAFLD group and the control group. **(C)** Correlation scatter plot between *IL6* and Neutrophils. **(D)** Correlation scatter plot between *IL6* and Eosinophils. Asterisks indicate p-values: ****p<0.0001, ***p<0.001, **p<0.01, *p<0.05.

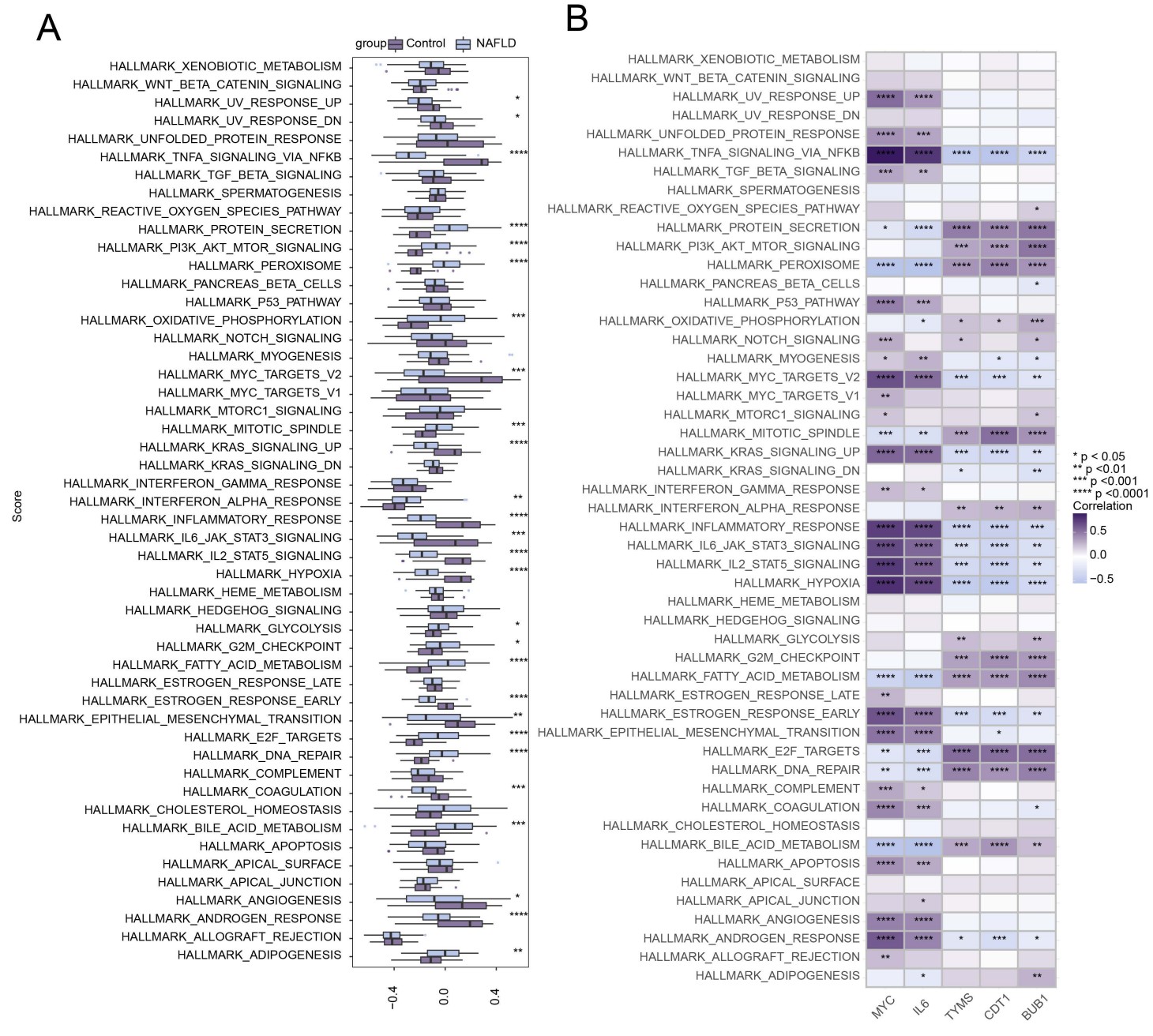

**Fig 11. Correlation between hub genes and 50 hallmark signaling pathways. (A)** Comparison of 50 Hallmark Signaling Pathways between the NAFLD Group and the Control Group. **(B)** Correlation Between Hub Genes and 50 Hallmark Signaling Pathways. ****$p < 0.0001$, ***$p < 0.001$, **$p < 0.01$, *$p < 0.05$.

injury and inflammation, while targeting iron metabolism (iron chelators) or antioxidant pathways (Nrf2 activators) shows synergistic effects [23,24]. Therefore, a deeper understanding of the role of ferroptosis in the pathogenesis of NAFLD is essential for developing effective diagnostic and therapeutic strategies.

Ferroptosis might play a key role in NAFLD's pathogenesis.Research on ferroptosis-related biomarkers mainly focuses on iron metabolism (Ferritin, Transferrin, Hepcidin), lipid peroxidation (ACSL4, GPX4, SLC7A11, etc.), lipoxygenases (such as ALOX15), lipid peroxidation products (MDA, 4-HNE), and antioxidant system markers (GSH, NRF2) [2]. Most of the existing biomarkers are systemic indicators, making it difficult to distinguish between hepatic and systemic metabolic abnormalities. For example, while serum ferritin (SF) is associated with hepatic iron deposition, its levels are also influenced by inflammation, insulin resistance, and obesity, lacking specific differentiation capability for the pathological stages of NAFLD (such as simple steatosis and NASH) [25]. Through bioinformatics analysis, we integrated gene expression profiling data and co-expression networks, identifying differentially expressed genes related to ferroptosis. These results provide new insights on the potential mechanisms of NAFLD and offer important directions for future clinical applications [4,5,26]. In this study, our gene expression analysis revealed 1,770 differentially expressed genes (DEGs), with 1,073 genes upregulated and 697 genes downregulated,the upregulated genes might be related to key biological processes such as lipid metabolism and inflammatory responses, while the downregulated genes might affect antioxidant capabilities and the regulation of iron metabolism [27].

Pathway analysis revealed significantly reduced expression in pathways related to folate metabolism and glycosaminoglycan biosynthesis in the NAFLD group, whereas expression related to non-homologous end joining and base excision repair pathways were significantly increased. These pathway changes might unveil potential mechanisms of NAFLD, suggesting a link between metabolic disturbances and disease progression. The lower expressed pathways might be associated with metabolic imbalances and weakened damage repair mechanisms, while the higher expressed pathways might reflect adaptive responses to cellular damage.

Simultaneously, we found a strong correlation between IL-6, neutrophils and eosinophils. IL-6, as a multifunctional cytokine, plays a dual role in NAFLD. The release of ferroptosis-induced IL-6 promotes the recruitment and activation of neutrophils, enhancing their capacity to release reactive ROS and inflammatory mediators, which exacerbates inflammation and oxidative stress in the liver and creats a vicious cycle that drives the progression of NAFLD to NASH and even cirrhosis [28]. On the contrary, IL-6 could modulate the function of eosinophils, promoting the release of anti-inflammatory cytokines such as IL-4 and IL-10, thereby inhibiting excessive inflammatory responses and alleviating liver damage and fibrosis to some extent [29]. However, the regulation of neutrophils and eosinophils by IL-6 driven by ferroptosis is complex and bidirectional, exhibiting both pro-inflammatory and anti-inflammatory effects. This balance plays a critical role in the pathogenesis of NAFLD. Further research into the specific molecular mechanisms of ferroptosis and IL-6 regulation of immune cells will not only contribute to a deeper understanding of the molecular pathology of NAFLD but also provide potential targets for developing therapeutic strategies targeting ferroptosis and IL-6 signaling pathways. For example, inhibiting ferroptosis or modulating the IL-6 signaling pathway might effectively mitigate inflammation and fibrosis in NAFLD by reducing neutrophil overactivation and enhancing the anti-inflammatory function of eosinophils [30].

The 41 intersecting genes extracted using the CytoHubba algorithm, including JUN, BRCA1, and EDN1, might play significant roles in the development and progression of NAFLD. The interactions of these genes provide insights into the molecular mechanisms of NAFLD and could potentially serve as biomarkers for targeted therapies. Finally, the diagnostic value of these intersecting genes was validated using ROC curves, with 25 genes exhibiting AUC values greater than 0.85, indicating strong discriminatory ability as potential biomarkers for NAFLD. Genes such as *MYC*, *SDC1*, and *FGFR2* might directly or indirectly influence ferroptosis by affecting pathways related to Nrf2 expression, glutathione synthesis, cell signaling, energy metabolism, and inflammatory signaling [31,32]. Overexpression of *SDC1* hinders insulin receptor activation and activates p53, a factor associated with ferroptosis [33–37]. FGFR2-mediated NF-κB activation sustains a glycolytic phenotype, affecting susceptibility to ferroptosis [38]. CRISPR-Cas9 gene editing indicates that *PGC1A* influences the expression of lipid metabolism-related genes [39]. As a rate-limiting enzyme in cholesterol catabolism, *CYP7A1* might lead to the accumulation of free radicals [40,]. Genes such as RGS1, IL2RB, IL6, and *CD3G* could be related to ferroptosis sensitivity within inflammatory signaling [41–45]. Genes like CDC20, TYMS, ASPM, CDT1, *MND1*, BUB1, and

*PBK* support hepatocyte proliferation, which is crucial for liver regeneration after injury [46–52]. *UBE2* and *ISG15* play essential roles in the ubiquitination pathway, impacting protein degradation and DNA repair [53]. Therefore, these genes might influence ferroptosis by directly or indirectly affecting pathways that regulate oxidative stress response and lipid metabolism.

This discovery provides essential insights for the early diagnosis and personalized treatment of NAFLD. Future research could focus on how these genes could be applied in clinical practice to improve the diagnostic accuracy and therapeutic efficacy for NAFLD. Additionally, further investigation of the intersecting genes should include their potential applications in other related diseases [54].This study reveals a unique regulatory network of ferroptosis in NAFLD through bioinformatics analysis, where the GO/KEGG enrichment results are similar to previous literature regarding metabolic regulatory pathways. However, the identification of certain hub genes (such as ASPM and CDC20) provides new perspectives on the mechanisms of NAFLD [55].

The limitations of this study primarily lie in several aspects. First, although we identified several differentially expressed genes associated with NAFLD using bioinformatics methods, the lack of wet lab validation might affect the reliability of the results. Second, the relatively small sample size might reduce the statistical power and limit the generalizability of the findings. Particularly, the sample size of control in GSE130970 dataset is small. Additionally, variability in the datasets could introduce batch effects, affecting the accuracy of the results. Therefore, future research should incorporate larger sample sizes and experimental validation to enhance the credibility of these findings.

In conclusion, by analyzing gene expression, pathway enrichment, and immune response in NAFLD, this study reveals potential molecular mechanisms and identifies a series of genes that could serve as biomarkers. These findings provide new perspectives on the early diagnosis of NAFLD and lay the groundwork for designing clinical intervention strategies. Future research should further validate these biomarkers' clinical application potential to advance personalized treatment for NAFLD.

## Author contributions

**Conceptualization:** Yongmei Zeng.

**Investigation:** Jiyong Zhang, Xiaoying Qiu, Yongmei Zeng.

**Methodology:** Jiyong Zhang, Weiyan Li, Xiaoying Qiu, Yuanyuan Wang, Yongmei Zeng.

**Project administration:** Weiyan Li, Xiaoying Qiu, Yuanyuan Wang, Yongmei Zeng.

**Resources:** Yuanyuan Wang.

**Software:** Weiyan Li.

**Validation:** Yongmei Zeng.

**Writing – original draft:** Jiyong Zhang.

**Writing – review & editing:** Jiyong Zhang, Yongmei Zeng.

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
