## [Decision Letter · Decision Letter 0]

21 Mar 2025

Dear Dr. Zeng,

We look forward to receiving your revised manuscript.

Kind regards,

Marwan Al-Nimer

Academic Editor

PLOS ONE

Journal Requirements:

https://blogs.plos.org/plos/2019/06/looking-good-tips-for-creating-your-plos-figures-graphics/

Additional Editor Comments:

The authors succeeded in identifying a new biomarkers that bridge the ferroptosis and NALFLD. Few points needed an explanation:

1: The new terminology for NAFLD is metabolic dysfunction associated liver disease (MASLD). Is changing the term will change your results? and are patients have diabetes mellitus or not?

2: Typing errors require correction e.g., in introduction, ofmechanisms, accuracyand........etc.

3:Introduction: Add a paragraph about the role ferroptosis in NAFLD or MASLD because the introduction is with few references.

4: The sample size of control in GSE130970 dataset is small. It can mentioned as a limitation of the study

5. It is a little be confused about the correlation test, is it Pearson or Spearman?

6. The area under the curve needs to be supplemented with probability and 95% confidence interval

7. Improving the quality of figures is important to be more attractive.

Reviewers' comments:

Reviewer's Responses to Questions

**Comments to the Author**

1. Is the manuscript technically sound, and do the data support the conclusions?

Reviewer #1: Yes

2. Has the statistical analysis been performed appropriately and rigorously?

Reviewer #1: Yes

3. Have the authors made all data underlying the findings in their manuscript fully available?

Reviewer #1: Yes

4. Is the manuscript presented in an intelligible fashion and written in standard English?

Reviewer #1: No

Reviewer #1: This dataset is commendably extensive for the NAFLD field; however, the manuscript’s writing still needs improvement, also, please follow the journal's format of publication. Please address the points outlined in the attached comments.

**Do you want your identity to be public for this peer review?** For information about this choice, including consent withdrawal, please see our Privacy Policy

Reviewer #1: No

---

## [Author Response · Author response to Decision Letter 1]

29 May 2025

Responses are attached in the file uploaded.

---

## [Editor Report · Decision Letter 1]

1 Jun 2025

Exploration of Key Markers Driving Ferroptosis in the Progression of Non-Alcoholic Fatty Liver Disease

PONE-D-25-07526R1

Dear Dr. Yongmei zeng

We’re pleased to inform you that your manuscript has been judged scientifically suitable for publication and will be formally accepted for publication once it meets all outstanding technical requirements.

Kind regards,

Marwan Al-Nimer

Academic Editor

PLOS ONE

Additional Editor Comments (optional):

No comments
---

## [Editor Report · Acceptance letter]

PONE-D-25-07526R1

PLOS ONE

Dear Dr. Zeng,

I'm pleased to inform you that your manuscript has been deemed suitable for publication in PLOS ONE. Congratulations! Your manuscript is now being handed over to our production team.

Kind regards,

on behalf of

Professor Marwan Salih Al-Nimer

Academic Editor

PLOS ONE